# A Laboratory Device Designed to Detect and Measure the Resistance Force of a Diagonal Conveyor Belt Plough

**DOI:** 10.3390/s23063137

**Published:** 2023-03-15

**Authors:** Leopold Hrabovský, Jakub Gaszek, Ladislav Kovář, Jiří Fries

**Affiliations:** Department of Machine and Industrial Design, Faculty of Mechanical Engineering, VSB-Technical University of Ostrava, 70800 Ostrava, Czech Republic

**Keywords:** conveyor belt diagonal plough, diversion plough, traction force measurement, motion resistance, shear friction coefficient, conveyor belt

## Abstract

This paper presents a laboratory device simulating a section of a conveyor belt on which a diagonal plough is installed. Experimental measurements were carried out in a laboratory belonging to the Department of Machine and Industrial Design at the VSB-Technical University of Ostrava. During the measurements, a plastic storage box, representing a piece load, transported on the surface of a conveyor belt at a constant speed was brought into contact with the front surface of a diagonal conveyor belt plough. The aim of this paper is to determine the amount of resistance generated by the diagonal conveyor belt plough when it is placed at different angles of inclination β [deg] in relation to the longitudinal axis, based on the experimental measurements performed using a laboratory measuring device. Based on the measured values of tensile force required to keep the conveyor belt moving at a constant speed, the resistance to the conveyor belt movement is expressed, with a value of 20.8 ± 0.3 N being attained. Based on the ratio of the measured value of the arithmetic average of the resistance force and the weight of the used length of the conveyor belt, a mean value of the specific movement resistance of the size 0.33 [N·N − 1] is calculated. This paper presents the time records obtained by measuring the tensile forces, on the basis of which it is possible to determine the magnitude of the force. The resistance during the ploughing operation of the diagonal plough when acting on a piece load placed on the working surface of the conveyor belt is presented. From the measured values of tensile forces presented in the tables, this paper reports the calculated values of the friction coefficient obtained during the movement of the diagonal plough when moving a piece of load with the defined weight from the working surface of the relevant conveyor belt. The maximum value of the arithmetic mean for the friction coefficient in motion µ = 0.86 was measured at an inclination angle of the diagonal plough of β = 30 deg.

## 1. Introduction

Conveyor belts [1] are classified as continuously operating transport equipment, and are used to move loose materials and piece materials over short and long distances. Loads are transported on the belt part of the conveyor belt, which is made of rubber [2,3], PVC [4] or steel [5,6]. These loads are fed via transfer points through the end (usually driven) drums of the conveyor belt, which are installed in the conveyor line in front of the conveyor belt, or by the feed of material flowing from the discharge hole of a hopper or by feeding material into the so-called filling point, which is a feed chute.

The material transported by the conveyor belt is removed from the conveyor belt either as a result of either it overflowing over the end drum or by pushing the material off at some point on the transport route using a V-plough (i.e., two-sided) [7] or a diagonal plough (i.e., single-sided) [8] scraper [9].

Loose material placed on a conveyor belt moves, due to the transmission capability of the drive (i.e., friction occurring in the contact surface of the driven drum shell and conveyor belt), with a constant velocity v [m·s^−1^], giving rise to the F_1_ [N] component in conveyor belts [10,11] for the motion resistance, which is called “conveyor belt length resistance“. The magnitude of this motion resistance component is directly proportional to the horizontal distance (length) L [m] of the conveyor belt.

Additional equipment for conveyor belts also includes so-called belt cleaners. The purpose of primary belt scrapers [12] or secondary belt scrapers [13] consisting of a blade or rotary brush belt cleaners [14] is to wipe the remaining grains of the transported material off from the working surface of the conveyor belt. Material grains that fall over the edges of the conveyor belt’s working surface and fall on the surface of the return branch (see Figure 1) are removed from the return branch of the conveyor belt using belt cleaners, either V-plough belt cleaners [15] or diagonal belt cleaners [16,17].

If we place a diverter plough at an angle of β [deg], then at that point, the transported material will be removed and ploughed over one or both edges of the conveyor belt. Ploughing the material produces an additional resistance force F_a_ [N], the magnitude of which can be expressed as shown in (1), see [10]. The magnitude of the additional resistance, which is represented by the resistance of the material plough F_a_ [N], is directly proportional to the conveyor belt width B [m] and the ploughing coefficient k_a_ [-] (k_a_ = 1500 N·m^−1^ [10]).
(1)$$F_{a}=B\cdot k_{a}\;[N]$$

According to [11], the ploughing coefficient k_a_ [-] should be determined as k_a_ = 1200 ÷ 1500 N·m^−1^, where smaller values are chosen for the lower widths B [m] of the conveyor belts.

From the analysis carried out by the authors of this paper, it can be concluded that there is currently not enough information—or articles providing information—on the magnitude of resistance forces produced by the scraper ploughs installed on conveyor belts.

Hrabovský et al. in their article [9] dealt with a theoretical expression of resistances occurring when pushing loose material off the conveyor belt surface with a single-sided plough. The working branch of a conveyor belt was designed in a flat, single-idler arrangement. Theoretical prerequisites and derivations were practically tested using a conveyor belt model.

Y. Nata et al. reported in [18] that the use of a V-plough is most frequently applied for conveyor belt transportation systems in coal fuel power plants, where the V-plough functions as a coal plough divider placed in the conveyor belt before entering the inlet chute. In this study, the relationship between the wide and long angles of a V-plough diverter, and the cleaning of conveyor belts in the conditions of the two dimensions, was presented.

In [19], V. V. Efremenkov examined the details of using diagonal ploughs in transport lines conveying batch and cullet to the hoppers of glass furnace chargers. An improved design for a diagonal plough equipped with an electromagnetic vibrator was introduced. An adaptive algorithm controlling the unloading of batch and cullet from a conveyor belt was presented. It was shown that the additional mixing of these materials on a belt was possible.

B. S. Moor, in the article [20], described the use of a curved plough used in mills located in South Africa supplying fuel to boilers from a conveyor. The curvature initiates bagasse movement along the plough at an acute angle of approach; by the stage at which the plough is square to the belt travel, the bagasse on the plough had already developed a component of momentum across the belt.

M. C. Makutu et al. stated in their article [21] that operating a plough scraper and a tilting device should be very easy and safe. Inasmuch as the installation and maintenance is cheaper, it does not require too much for both operation and maintenance. Some systems can be automated with the plant, and require no operator for manual operation.

The aim of this paper, based on experimental measurements performed on a laboratory machine, is to detect the resistance forces that are generated when ploughing a piece load from the surface of the conveyor belt using a one-sided scraping plough. The measured resistance forces (when ploughing a piece load from the working surface of a conveyor belt with additional equipment fixed to it, i.e., the so-called diagonal plough) were determined for six angles β = 0÷50 deg of the belt scraping plough. A piece load (consisting of a plastic crate with weights of different masses) was transported on the surface of the conveyor belt at a constant velocity, and during its contact with the scraping plough, the values of the force required to maintain the constant speed of the conveyor belt movement were measured using a load sensor. From the measured values and results, the mean resistance values of the scraping plough were determined, as was the coefficient of friction from the contact surface of the load and the conveyor belt.

## 2. Materials and Methods

A laboratory machine (Figure 2) was used for experimental measurements, simulating a driven conveyor belt with an installed diagonal plough, which was designed and modeled using SolidWorks^®^Premium 2012 SP5.0 software at the Department of Machine and Industrial Design, Faculty of Mechanical Engineering, VSB-Technical University of Ostrava.

The laboratory testing device consists of a steel structure to which a drive drum is mechanically fixed 1. The drive drum 1, marked by its producer as LAT216.1 [22] (with technical parameters n_b_ = 32 min^−1^, M_2_ = 441 N·m, F_b_ = 4090 N), was purchased from the company Bluetech s.r.o., Pacov, Czech Republic. This drum with a diameter of D_b_ = 216 mm is made of steel, and rotated at a peripheral speed of v_b_ = 0.37 m·s^−1^ in bearings pressed on a shaft with a diameter of 40 mm. The drive drum, with a built-in planetary gearbox (a gear ratio i_p_ = 29.06), uses a three-phase electric motor with the power of P_e_ = 1.5 kW and nominal speed n_e_ = 930·min^−1^.

One end of steel rope 2 (with a diameter of 2 mm) is wound on the drive drum 1, which is attached to the steel structure of the laboratory device and has a nominal speed of n_b_ [s^−1^] controlled by a frequency converter 8 (type: YASKAWA VS-606 V7 [23]) and its size is detected using the tachometer UNI-T UT373 9 [24]. The other end of rope 2 is fixed to a strain gauge load cell S-AST 3 [25], which is mechanically attached to the conveyor belt 4 (width B = 500 mm, length L_p_ = 1650 mm, construction EP250/2). Winding the free end of the steel rope 2, which is attached to the strain gauge load cell 3 by its other end, to the drive drum 1 rotating at the speed (revolutions n_s_ [s^−1^]) sets the conveyor belt 4 in motion, and reaches the speed v_s_ [m·s^−1^].

The beginning of the conveyor belt 4, supported by transition idlers 5 (diameter 63 mm, length 500 mm) was moved to the required distance from the axis of the drive drum 1 before each experimental measurement to guarantee a sufficient distance for the front surface of the conveyor belt 4 to the drive drum 1 during the laboratory measurements.

To the structure of this laboratory device, the diagonal plough 6 was mechanically fixed using bolted joints.

After positioning the conveyor belt 4 to the right position, and after setting the scraper plough 6 to the desired position (defined by angle β [deg]), the electric motor of drive drum 1 was connected to the mains. By rotating the drive drum 1 at the speed of n_s_ [s^−1^], the steel rope 2 was wound on the drive drum casing at speed vs. [m·s^−1^], the magnitude of which can be expressed using Relationship (2).
(2)$$v_{s}{=D}_{b}\cdot \pi \cdot n_{s}{=k}_{\mathrm{FM}}\cdot D_{b}\cdot \pi \cdot n_{b}\;[m\cdot s^{-1}]$$
where k_FM_ [-] is the constant expressing the value of the pre-set frequency of the oscillation of the frequency converter 8 (k_FM_ = 0 ÷ 1).

The first experimental measurements were carried out in order to obtain the value of the arithmetic mean of the tensile force F_M1_ [N], which expresses the resistance to the movement of an unloaded conveyor belt (G_b_ = 0 N) on the transport idler of this laboratory device (see Figure 2).

Subsequent experimental measurements were carried out to obtain the values of measured tensile forces F_M2(i)_ − F_M1(i)_ [N], which express the resistance while the material was being ploughed (from a conveyor belt loaded with a load G_b_ [N]) using a diagonal plough. On the working surface of the conveyor belt 4 (in front of the scraper plough 6), a piece of load 7 of the known weight G_b_ [N] is placed. The piece load consists of a plastic crate with dimensions 400 × 300 × 100 mm and a weight of 20 N. By rotating the drive drum 1 with the rotation n_s_ [s^−1^] the steel rope 2 was wound onto the drive drum casing at a speed of vs. [m·s^−1^], the magnitude of which can be expressed as shown in Relationship (2), and is identical to the speed of conveyor belt movement 4.

Experimental measurements were used to obtain the values of resistance difference forces F_M2(i)_ − F_M1(i)_ [N], which a load 7 with the weight G_b(i)_ [N] will impose when ploughing it down using a scraper plough 6, inclined at an angle β = 0 ÷ 50 deg perpendicular to the longitudinal axis of the conveyor belt 4.

The time of ploughing the piece load from the conveyor belt by a scraper plough is divided into three phases. In the first phase, the edge of the piece load comes into contact with the surface of the scraper plough, see Figure 3a. At this point, there is a speed reduction (at β > 0 deg) or a stoppage (at β = 0 deg) of the load carried by a conveyor belt, which moves at a speed equal to that of the conveyor belt. When the conveyor belt moves, a resistive force F_M2(i)_ − F_M1(i)_ [N] is generated (the resistance of the material scraper plough), which can be theoretically expressed as a frictional force T_a_ [N], see (3), where μ [-] is the coefficient of shear friction on the contact surface between the load and the conveyor belt surface.
(3)$$F_{M2}{\;-\;F}_{M1}{=T}_{a}{=G}_{b}\cdot \mu \;[N]$$

In the second phase, the piece load on the conveyor belt is rotated until the moment when the entire front surface of the piece load is aligned with the surface of the scraper plough, see Figure 3b. When the conveyor belt moves, a resistive force F_M2(i)_ − F_M1(i)_ [N] (the resistance of the material scraping plough) is generated, which can theoretically be expressed as a frictional force T_b_ [N], see (4), where f [-] is the coefficient of shear friction during movement on the contact surface of the load and the surface of the conveyor belt.
(4)$$F_{M2}{\;-\;F}_{M1}{=T}_{b}{=G}_{b}\cdot f\;[N]$$

In the third stage, the piece load is moved along the surface of the conveyor belt. When the conveyor belt moves, a resistive force F_M2(i)_ − F_M1(i)_ [N] is generated, which can theoretically be expressed as the sum of the frictional force Tb [N] and Tc [N], see (5), where f1 [-] is the coefficient of shear friction in motion on the contact surface of the load and the surface of the scrap shield, and F [N] is the vertical load force acting on the front surface of the scraper plough.
(5)$$F_{M2}{\;-\;F}_{M1}{=T}_{b}{+T}_{c}{=G}_{b}\cdot f+F\cdot f_{1}\;[N]$$

The experimental measurements aimed to measure:The magnitude of tensile force F_M1(i)_ [N] needed to carry the conveyor belt of a weight G_p_ = 63.3 N at a constant speed vs. [m·s^−1^] on the idle rollers of the laboratory device,The magnitude of tensile force F_M2(i)_ − F_M1(i)_ [N] necessary to plough the piece load of weight G_b_ [N] from the surface of the conveyor belt when the diagonal plough is inclined at 0 deg, 10 deg, 20 deg, 30 deg, 40 deg and 50 deg. The load consisted of a plastic crate (with a weight of 20 N) and a maximum of six pieces of weight, where one piece of weight had a weight of 5 kg.

The measuring chain applied to the experimental measurements of tensile forces F_M1(i)_ [N] and F_M2(i)_ [N] when the load is ploughed from the conveyor belt travelling with a constant speed vs. [m·s^−1^] on the surface of which the load is carried away, is displayed in Figure 4.

A load sensor cable 5 equipped with a D-Sub plug was plugged into the socket of the measuring module BR4D [26] of the strain gauge apparatus DS NET [26] during the laboratory measurements. A laptop was connected to the DS NET strain gauge using a network cable with RJ-45 connectors at both ends.

## 3. Results

The measurements obtained on a laboratory machine, see Figure 2, were performed in order to fulfil two objectives. The first was to record a sufficient number of the measured values of the tensile forces that are caused by conveyor rollers (i.e., motion resistance) in relation to the moving conveyor belt with the final length of L_p_ = 1.65 m, which they support. In this case, the tensile forces were detected only for vertical loads, the magnitude of which was defined by the weight of the conveyor belt G_p_ = 63.3 N. For details, see Section 3.1.

The second objective was to obtain a sufficient number of the measured values of the tensile forces that are generated during the contact of the piece load by gravity Gb [N] (carried away on a conveyor belt of the length Lp = 1.6 m at a constant speed of vs. [m·s^−1^]) with a scraper plough inclined at an angle β = 0÷50 deg perpendicular to the longitudinal axis of the conveyor belt. For more details, see Section 3.2 to Section 3.7.

### 3.1. The Measurement of Resistance Forces Acting against the Conveyor Belt without a Load

The conveyor belt 4 with the final length L_p_ = 1.65 m, connected mechanically in the front part to load cell 3, was set in motion as a result of rope 2 being wound at a speed vs. [m·s^−1^] on the drive drum 1. The load sensor 3 during the period of rope winding 2 on the drum 1 detected the values of tensile force F_M1(i)_ [N], generated by a moving conveyor belt 1 supported by the conveyor idlers 5. The values of tensile force F_M1(i)_ [N] for the number of measurements i = 10 were recorded in Table 1.

Using the ratio of the measured resistance force F_M1(i)_ [N] value and the weight of the conveyor belt G_p_ [N], the values of the drag coefficient w_(i)_ [-] were calculated.

In Table 1, the calculated arithmetic means of all i = 10 measured for the force F_M1(i)_ [N] are given, as well as the calculated values of the drag coefficient w_(i)_ [-], including extreme measurement errors κ_α,i_ [N], for the selected risk of Student distribution α = 5% and Student coefficient t_α,i_ = 2.26.

### 3.2. The Measurement of Movement Resistance Forces of a Loaded Conveyor Belt at the Scraper Plough Inclination of β = 0 deg

Table 2 shows the instantaneous values (for measurements repeated three times, i.e., i = 3) of the tensile forces F_M1(i)_ [N] and F_M2(i)_ [N], read from the PC display (software DEWESoft X2 SP5) for different load weights G_b(i)_ [N] at the angle of inclination for the scraper plough β = 0 deg.

On the basis of i = 3 repeated measurements under the same conditions, the values of pulling forces F_M2(i)_ − F_M1(i)_ [N] were measured. In the critical values table of the Student distribution [27], for the selected risk α = 5%, the Student coefficient was t_α,i_ [-] [27]. According to [27], the standard deviation of the arithmetic mean s_o_ [N] was calculated for i = 3 repeated measurements. Three standard deviations κ_α,i_ [N], see the last row of Table 1 and other tables displayed in this section, were calculated as the product of t_α,i_·s_o_.

Figure 5 presents selected tensile force courses, see Table 2, which apply to a scraper plough inclined perpendicular to the longitudinal axis of the conveyor belt at an angle β = 0 deg.

The measured courses of tensile forces, listed in Table 2 (as well as the other tables in Section 3), not listed in Figure 5 (and also in other figures in Section 3) have been archived by the authors of this article. By prior arrangement, they can be provided to potentially interested parties.

### 3.3. The Measurement of Resistance Forces Acting against the Movement of the Loaded Conveyor Belt at the Angle of Scraper Plough Inclination β = 10 deg

Table 3 shows the instantaneous values of the measured tensile forces F_M1(i)_ [N] and F_M2(i)_ [N], read from DEWESoft X2 SP5 software for different load weights G_b(i)_ [N] at the scraper plough inclination β = 10 deg.

Figure 6 presents selected tensile force courses, see Table 3, which apply to a scraper plough inclined perpendicular to the longitudinal axis of the conveyor belt at an angle β = 10 deg.

### 3.4. The Measurement of Resistance Forces Acting against the Movement of the Loaded Conveyor Belt at the Angle of Scraper Plough Inclination β = 20 deg

Table 4 shows the instantaneous values of the measured tensile forces F_M1(i)_ [N] and F_M2(i)_ [N], read from DEWESoft X2 SP5 software for different load weights G_b(i)_ [N] at the scraper plough inclination β = 20 deg.

Figure 7 presents selected tensile force courses, see Table 4, which apply to a scraper plough inclined perpendicular to the longitudinal axis of the conveyor belt at an angle β = 20 deg.

### 3.5. The Measurement of Resistance Forces Acting against the Movement of the Loaded Conveyor Belt at the Angle of Scraper Plough Inclination β = 30 deg

Table 5 shows the instantaneous values of the measured tensile forces F_M1(i)_ [N] and F_M2(i)_ [N], read from DEWESoft X2 SP5 software for different load weights G_b(i)_ [N] at the scraper plough inclination β = 30 deg.

Figure 8 presents selected tensile force courses, see Table 5, which apply to a scraper plough inclined perpendicular to the longitudinal axis of the conveyor belt at an angle β = 30 deg.

### 3.6. The Measurement of Resistance Forces Acting against the Movement of the Loaded Conveyor Belt at the Angle of Scraper Plough Inclination β = 40 deg

Table 6 shows the instantaneous values of the measured tensile forces F_M1(i)_ [N] and F_M2(i)_ [N], read from DEWESoft X2 SP5 software for different load weights G_b(i)_ [N] at the scraper plough inclination β = 40 deg.

Figure 9 presents selected tensile force courses, see Table 6, that apply to a scraper plough inclined perpendicular to the longitudinal axis of the conveyor belt at an angle β = 40 deg.

### 3.7. The Measurement of Resistance Forces Acting against the Movement of the Loaded Conveyor Belt at the Angle of Scraper Plough Inclination β = 50 deg

Table 7 shows the instantaneous values of the measured tensile forces F_M1(i)_ [N] and F_M2(i)_ [N], read from DEWESoft X2 SP5 software for different load weights G_b(i)_ [N] at the scraper plough inclination β = 50 deg.

Figure 10 presents selected tensile force courses, see Table 7, which apply to a scraper plough inclined perpendicular to the longitudinal axis of the conveyor belt at an angle β = 50 deg.

Figure 11 presents one of the ten measurements recorded using DEWESoft X2 SP5 software, from which the magnitude of the tensile force F_M1(i)_ [N] was determined, as listed in Table 1. None of the time recordings of the measured tensile forces F_M1(i)_ [N] obtained with the help of the laboratory device and DEWESoft X2 SP5 software are documented in this paper. (All i = 10 records are available to interested parties upon request from the authors of the article).

A part of the time curve for the measured force recording shown in Figure 12 is divided using dashed lines into three areas.

Table 2 to Table 7 show the magnitudes of the measured tensile forces F_M1(i)_ [N], needed to move the belt from a standstill state to motion at a constant speed, and also the magnitude of the measured tensile forces F_M2(i)_ [N], see Figure 13, generated in the course of ploughing a piece load from the working surface of the conveyor belt. For i = 3 calculated tensile values F_M2(i)_ − F_M1(i)_ [N] the arithmetic mean of the forces F_a(i)_ [N] was calculated using the Student distribution and standard deviation κ_5%,3_ [N], which define the resistance when ploughing a piece load by a scraper plough F_a_ [N] (1).

## 4. Discussion

The total movement resistance of a conveyor belt consists of individual resistances that, according to [10], can be classified into the following five groups: main resistances, secondary resistances, additional main resistances, additional secondary resistances and resistance to overcome the conveying height.

The group of additional secondary resistances of the conveyor belt includes five resistances, one of which is denoted as “resistance F_a_ [N] of the scrapers for a conveyed mass”, and its size can be calculated according to Relationship (1). It can be stated that Relationship (1) does not take into account the mechanical–physical properties of the transported materials, the conveyor belt speed, the conditions of the conveyor belt working surface, or the angle of inclination for the scraper plough.

For a conveyor belt with a width B = 0.5 m, according to [10], the resistance of the scraper for the conveyed material reaches a magnitude of F_a_ = 750 N.

Table 1 lists the magnitudes of the measured tensile forces F_M1(i)_ [N] needed to set the belt from a stationary state to a constant speed motion. The magnitude of each measured force F_M1(i)_ [N] can be defined from the momentum change theorem, i.e., that the driving force impulse is equal to the time change in the momentum change at a mass point. For i = 10 of the measured magnitudes of tensile forces needed to set the conveyor belt in motion, F_M1(i)_ [N], using the Student distribution, the arithmetic mean of forces F_M1_ = 20.8 N and the standard deviation κ_5%,10_ = 0.3 N were calculated.

Figure 11 shows three areas bounded by dashed lines.

The area A, marked in red, displays the time record of the tensile force F_M1(i)_ [N] measured at the moment when the conveyor belt of weight G_p_ [N] is supported by the conveyor idlers and is at its standstill position, e.g., it does not move at speed vs. [m·s^−1^]. The wavy part of the measured force curve F_M1(i)_ [N] presents the state of loose rope 2 winding on the drive drum 1, see Figure 2.

In the area B, marked in blue a yellow circle indicates the moment at which the tensile force F_M1(i)_ [N] overcomes the resistive force of the rope wound on the drive drum, imposed by the bearings of the conveyor idlers against the rotation of the casings of these idlers, supporting the conveyor belt of a length L_p_ = 1.65 m. At this point, the conveyor belt begins to move from its resting position.

The green area, C, represents a uniform rectilinear movement of the conveyor belt at speed vs. [m·s^−1^]. This is the area in which the values of the generated tensile forces F_M1(i)_ [N] were measured, as they were detected by single point load cells for the known weight of the conveyor belt G_p_ [N]. These are listed in Table 1.

Using the known values of the measured tensile forces F_M1(i)_ [N] and weight G_p_ = 63.3 N for the conveyor belt with the length L_p_ = 1.65 m used for the laboratory device, see Figure 2, the magnitudes of the specific motion resistance w_(i)_ [-] were calculated, see Table 1. Values calculated using the Student distribution, the arithmetic mean of the specific motion resistance w = 033, and the standard deviation κ_5%,10_ = 0.01 are displayed in Table 1.

In the red area, A, the time recording is shown for the measured tensile force F_M1(i)_ [N] at the moment when a piece load (the plastic crate of a weight 20.N with one to seven pieces, where one piece weighs 5 kg) reaches the weight G_b_ [N] is carried on the surface of the conveyor belt moving at a constant speed vs. [m·s^−1^]. Area A is described in detail in the comments for Figure 11.

In the area B, marked in blue, a blue ring indicates the moment at which static friction is overcome on the contact surface of the piece load and the surface of the conveyor belt. At this point, the piece load begins to slip on the surface of the conveyor belt moving at a constant speed vs. [m·s^−1^].

The green-marked area, C, represents a uniform rectilinear slipping of the piece load on the surface of the conveyor belt of the laboratory machine. In this area, the values of the generated tensile force F_M2(i)_ [N] were measured, detected by single point load cells, for the known load weight G_b_ [N], recorded in Table 2 to Table 7.

The maximum value of the arithmetic mean for the force F_a(i)_ = 243.9 ± 42.9 N was calculated for the load weight G_b_ = 296.2 N and a scraper plough at an inclination β = 10 deg, see Table 3. The resistance when ploughing the load reached a maximum value of F_a,max_ = (F_a(i)_ + κ_5%,3_)·B = (243.9 + 42.9)·0.5 = 143.4 N, which is 19.2% of the value of F_a_ = 750 N, calculated according to Relationship (1).

By the ratio of the arithmetic mean values of the measured forces F_a(i)_ [N] and the weight of the load G_b(i)_ [N], see Table 2 to Table 7, the coefficient of the friction in motion μ_(i)_ [-] was calculated for the loaded plastic crate against the rubber surface of the conveyor belt. The maximum value of the arithmetic mean of the friction coefficient in motion μ = 0.86 was measured at an angle of plough inclination β = 30 deg, see Table 5.

If loose material with a grain size defined according to the standard [11] is transported by a conveyor belt that has a specific weight 2100 kg·m^−3^, then the permissible load on the conveyor idlers results from the operating conditions, i.e., belt width, specific weight, the granularity of the transported material and the spacing of rollers. The loading of the middle idler for the conveyor belt at its full load of the conveyor belt with a width B = 0.5 m and a roller pitch of 1 m is recommended to provide m = 28 kg according to [27], and a weight of loose material reaching G = 274.7 N. If the friction coefficient acts on the contact surface of the ploughed grains of the loose material and the conveyor belt area (the maximum value chosen, calculated as the arithmetic mean of the calculated values µ_(i)_ [-], see Table 5) µ = 0.86, then the friction force [9] derived when the loose material is ploughed from the surface of the conveyor belt reaches T = m·g·µ = 28·g·0.86 = 236.2 N.

## 5. Conclusions

This paper presents a laboratory device, structurally designed and built in the Laboratory for Research and Testing at the Department of Machine and Industrial Design, Faculty of Mechanical Engineering, VSB-Technical University of Ostrava.

This laboratory device enabled us to measure the magnitude of tensile force generated as a resistive force when trying to set the non-moving conveyor belt in motion, and also when it moves at a constant speed on the conveyor idlers that support this belt.

One end of a steel rope was wound onto the drive drum and the other end was mechanically attached to the load cell, and the front area of the conveyor belt produced a tensile force. When the tensile force reached the magnitude of the force needed to spin the idlers supporting the conveyor belt, the moment comes at which the conveyor belt is set in motion from the standstill position. The time recording of the measured tensile force obtained using DEWESoft X2 SP5 software makes it possible to determine the instantaneous value of the tensile force, at any time during the recording of measurements, during the start-up and movement of the conveyor belt.

Based on the ratio of the known value for the measured tensile force with the uniform movement of the conveyor belt and the known weight of the load, the specific motion resistance can be calculated, see Table 1, which defines the energy performance of smoothly operating transport equipment, among which the conveyor belt is also included.

The laboratory machine was used to detect one of the five components of the additional side resistances of the conveyor belt, denoted as the “resistance of ploughs scraping the transported mass from the belt”. According to the standard [10], the magnitude of resistance for scraper ploughs designed to plough the transported mass should be calculated using Equation (1).

In this paper, the values obtained by measuring the tensile forces are presented in Section 3.2 to Section 3.7, which present the real values of resistance when ploughing a piece load from a conveyor belt using a scraper plough consisting of aluminum alloys.

Table 2 shows the measured resistance forces generated by a piece load carried on the surface at a constant conveyor belt speed, in contact with a scraper plough installed perpendicular to the longitudinal axis of the conveyor belt. The resistance forces are the frictional forces that act on the contact surface of the lower surface of the piece load (plastic crates with one to six pieces acting as weights) and the rubber surface of the conveyor belt. Based on the measured values of the tensile forces given and the known weight of the load, the specific motion resistance was calculated, as shown in Table 2, and this table also shows the values of the coefficient of shear friction during the movement of a plastic crate loaded with weights, skidding on the rubber surface of the conveyor belt.

Table 3, Table 4, Table 5, Table 6 and Table 7 give the calculated values of the shear friction coefficient in the motion of a plastic container loaded with weights, skidding on the rubber surface of the conveyor belt.

By determining the exact value of the total resistance when ploughing the piece material or loose material using a scraping plough conveyor belton a conveyor belt, it is possible to determine the sum of the two resistive forces [9]. The first resistive force, which reaches a higher value than the second force, is the component of the friction force that manifests itself when the material slides on the surface of the conveyor belt. The second resistive force is the frictional force acting when the material slides along the belt surface in contact with the surface of the scraper plough.

The unique contribution of this paper and, at the same time, a conveyor of new knowledge are the data presented in the tables for the listed values of the measured tensile forces. These are needed to keep the conveyor belt moving for the different weights of the piece loads placed on the surface of the conveyor belt, and which are in contact with the front surface of the scraper plough.

## Figures and Tables

**Figure 1 sensors-23-03137-f001:**
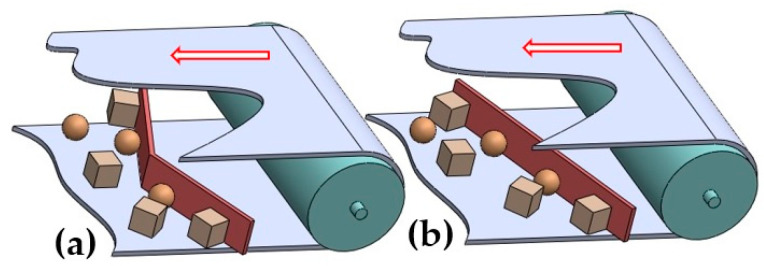
Belt cleaners: (**a**) V-plough, (**b**) diagonal.

**Figure 2 sensors-23-03137-f002:**
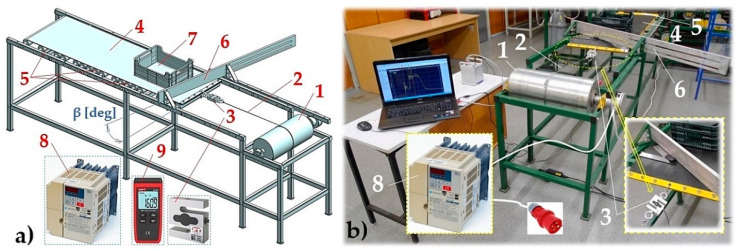
(**a**) 3D model of the laboratory device, used for experimental resistance measurements generated by a diagonal plough, created in the SolidWorks 2012 × 64 SP5.0 environment; (**b**) building the laboratory machine: 1—drive drum; 2—steel rope; 3—load cell; 4—conveyor belt; 5—conveyor idlers; 6—diagonal scraper plough; 7—plastic crate; 8—frequency converter; 9—tachometer.

**Figure 3 sensors-23-03137-f003:**
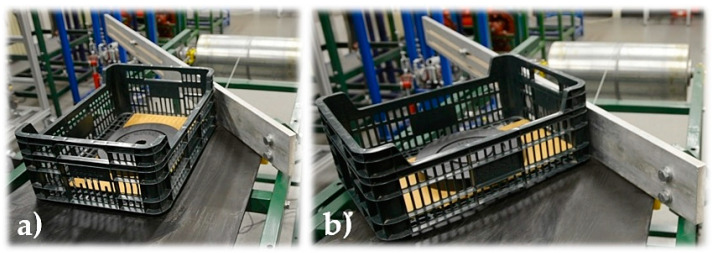
The position of the piece load (plastic crate) during its contact with the scraper plough at the moment of (**a**) phase 1 and beginning of the phase 2, (**b**) at the end of phase 2 and in the course of phase 3.

**Figure 4 sensors-23-03137-f004:**
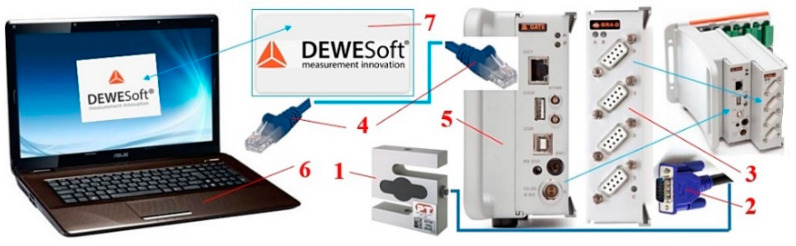
Measuring chain for the experimental measurement of conveyor belt tensile force: 1—load sensor; 2—D-sub plug; 3—measuring module BR4D of the strain gauge apparatus DS NET; 4—network cable; 5—gateway module DS GATE; 6—laptop; 7—software DEWESoft X2 SP5.

**Figure 5 sensors-23-03137-f005:**
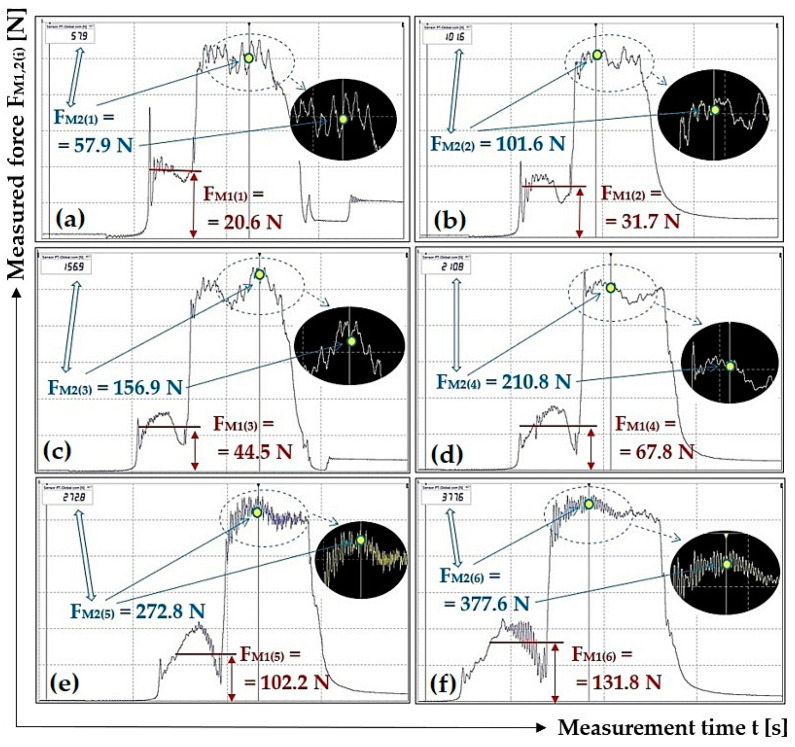
Tensile forces measured using DeweSoft^®^ software for different load weights at the scraper plough angle of inclination β = 0 deg.

**Figure 6 sensors-23-03137-f006:**
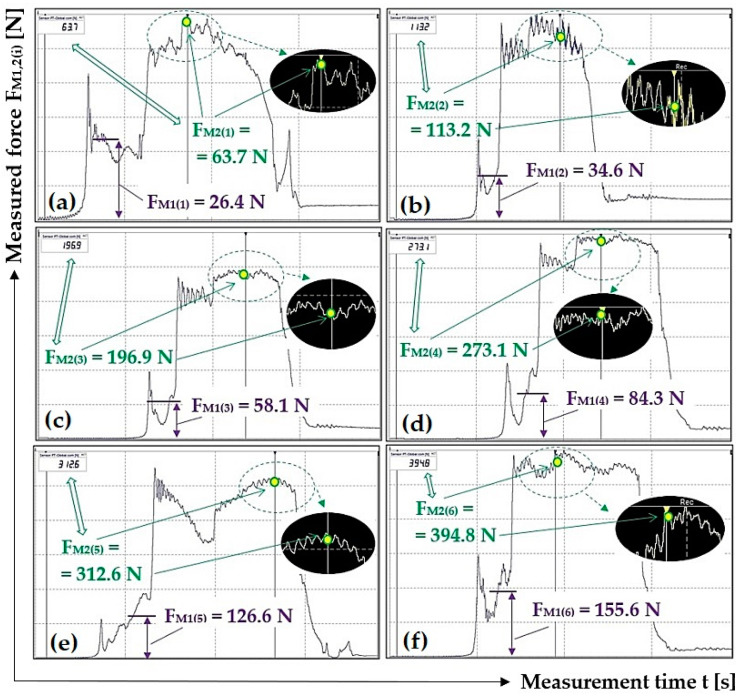
Tensile forces measured using DeweSoft^®^ software for different load weights at the scraper plough angle of inclination β = 10 deg.

**Figure 7 sensors-23-03137-f007:**
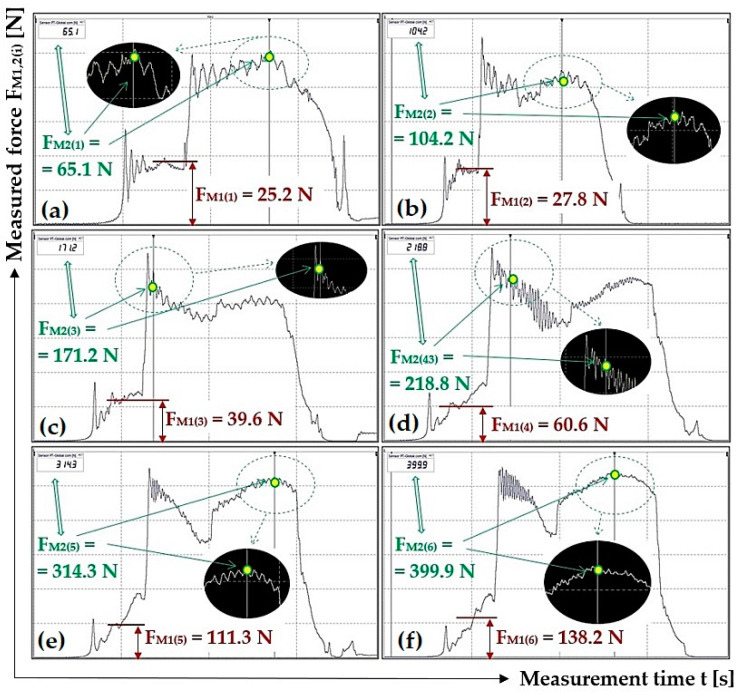
Tensile forces measured using DeweSoft^®^ software for different load weights at the scraper plough angle of inclination β = 20 deg.

**Figure 8 sensors-23-03137-f008:**
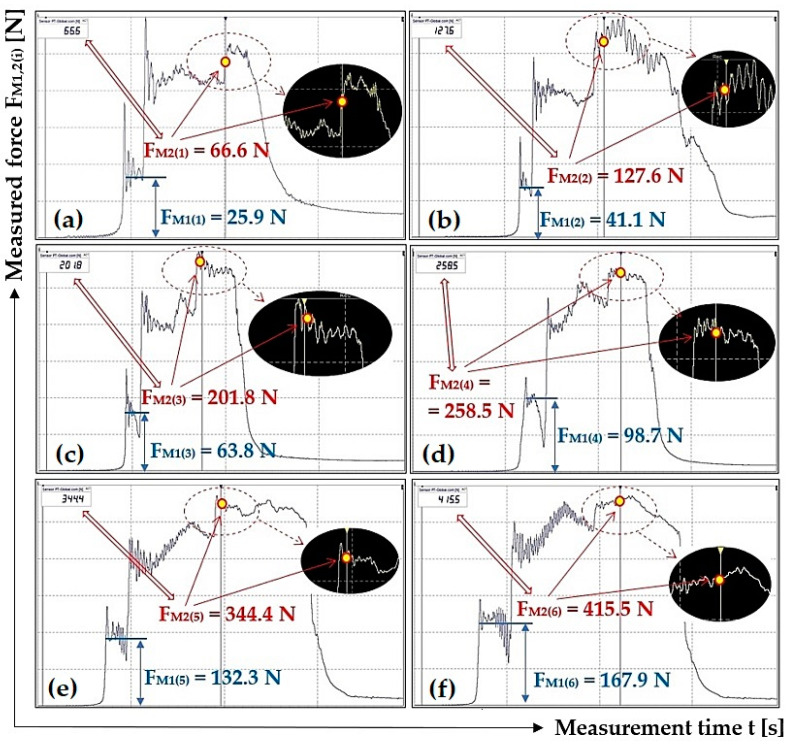
Tensile forces measured using DeweSoft^®^ software for different load weights at the scraper plough angle of inclination β = 30 deg.

**Figure 9 sensors-23-03137-f009:**
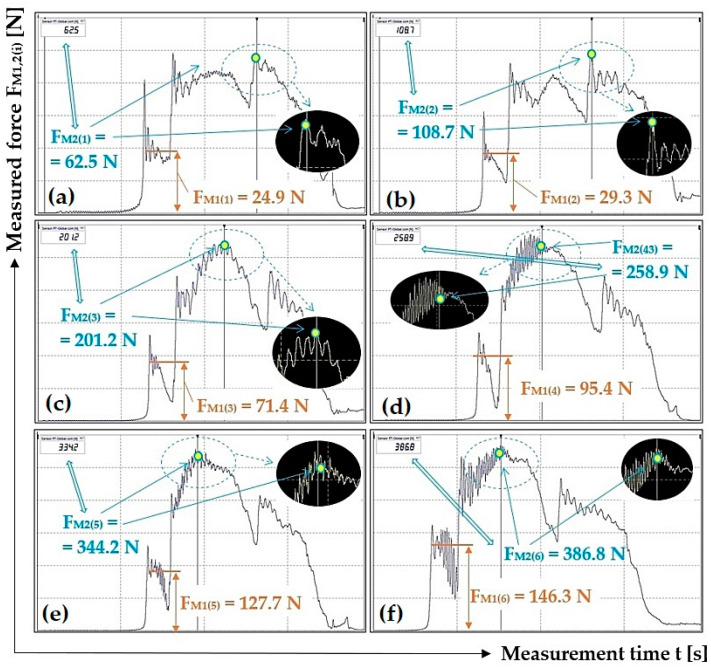
Tensile forces measured using DeweSoft^®^ software for different load weights at the scraper plough angle of inclination β = 40 deg.

**Figure 10 sensors-23-03137-f010:**
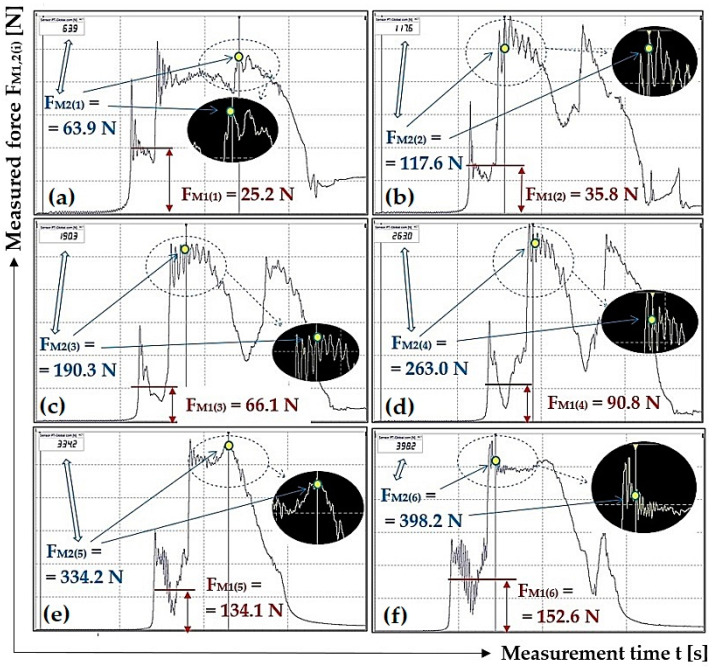
Tensile forces measured using DeweSoft^®^ software for different load weights at the scraper plough angle of inclination β = 50 deg.

**Figure 11 sensors-23-03137-f011:**
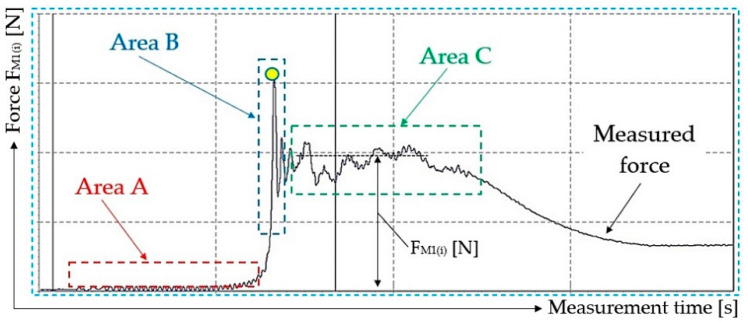
Tensile force recording, expressing the movement resistance of the conveyor belt.

**Figure 12 sensors-23-03137-f012:**
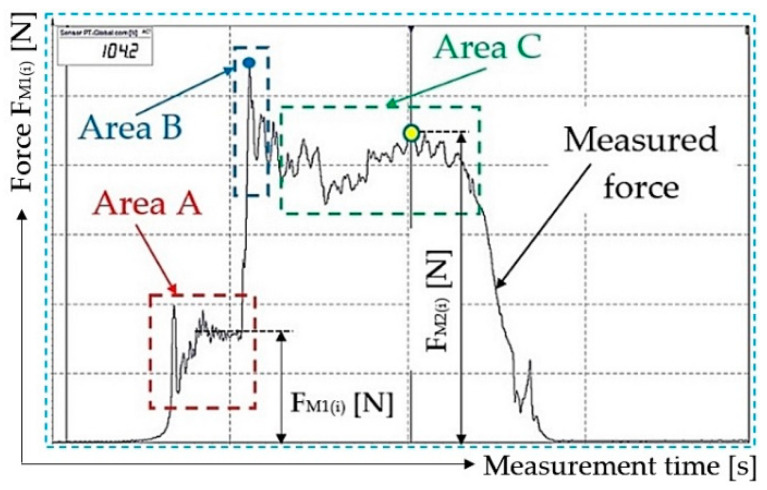
Tensile force measurement that defines the resistance of the scraper plough.

**Figure 13 sensors-23-03137-f013:**
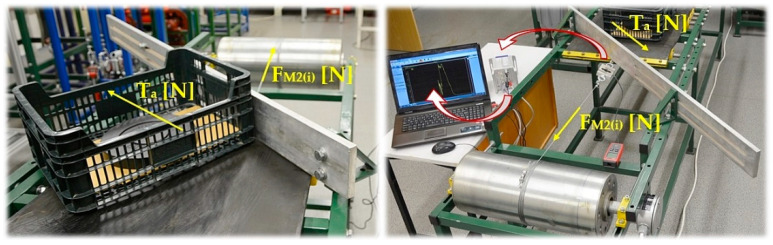
Tensile force F_M2(i)_ [N] measurement carried on using a laboratory device.

**Table 1 sensors-23-03137-t001:** Measured force F_M1(i)_ [N] for the weight G_p_ = 63.3 N acting on the conveyor belt with a length of L_p_ = 1.65 m.

**G_p_ [N]**	63.3											
**F_M1(i)_ [N]**	20.7	20.3	21.1	20.2	21.5	20.7	20.8	20.4	21.1	20.9	F_M1_ ± κ_5%,10_ [N]	20.8 ± 0.3
**w_(i)_ [-]**	0.33	0.32	0.33	0.32	0.34	0.33	0.33	0.32	0.33	0.33	w ± κ_5%,10_ [-]	0.33 ± 0.01

**Table 2 sensors-23-03137-t002:** Measured force F_M1(i)_ [N] and F_M2(i)_ [N] for the weight of the load G_b(i)_ [N] at angle β = 0 deg.

**G_b(i)_ [N]**	51	100.1	149.1	198.1	247.2	296.2
**F_M1(i)_ [N]**	20.6 *^1a^	31.7 *^1b^	44.5 *^1c^	67.8 *^1d^	102.2 *^1e^	131.8 *^1f^
	23.1	32.6	46.4	72.3	104.1	127.1
	23.8	33.2	46.8	75.2	105.5	135.6
**F_M2(i)_ [N]**	57.9 *^1a^	101.6 *^1b^	156.9 *^1c^	210.8 *^1d^	272.8 *^1e^	377.6 *^1f^
	60.6	103.3	160.1	217.4	281.6	384.6
	61.4	106.9	161.1	218.8	277.8	376.2
**F_M2(i)_ − F_M1(i)_ [N]**	37.3	69.9	112.4	143.0	170.6	245.8
	36.8	70.1	113.3	142.2	176.1	249.0
	37.6	73.7	114.3	143.6	172.3	240.6
**F_a(i)_ ± κ_5%,3_ [N]**	37.2 ± 1.1	71.2 ± 6.3	113.3 ± 2.4	142.9 ± 1.9	173.0 ± 7.9	245.1 ± 11.5
**μ_(i)_ [-]**	0.73	0.71	0.76	0.72	0.70	0.83

*^1a^ see Figure 5a, *^1b^ see Figure 5b, *^1c^ see Figure 5c, *^1d^ see Figure 5d, *^1e^ see Figure 5e, *^1f^ see Figure 5f.

**Table 3 sensors-23-03137-t003:** Measured force F_M1(i)_ [N] and F_M2(i)_ [N] for the weight of the load G_b(i)_ [N] at angle β = 10 deg.

**G_b(i)_ [N]**	51	100.1	149.1	198.1	247.2	296.2
**F_M1(i)_ [N]**	25.3	35.1	54.6	86.8	111.3	151.5
	26.4 *^2a^	34.6 *^2b^	58.1 *^2c^	84.3 *^2d^	126.6 *^2e^	155.6 *^2f^
	25.4	35.2	56.8	86.7	131.7	144.8
**F_M2(i)_ [N]**	63.4	109.4	176.1	258.2	314.8	383.1
	63.7 ^2a^	113.2 *^2b^	196.9 *^2c^	273.1 *^2d^	312.6 *^2e^	394.8 *^2f^
	64.9	106.7	177.6	249.6	328.9	415.6
**F_M2(i)_ − F_M1(i)_ [N]**	38.1	74.3	142.3	171.4	203.5	231.6
	37.3	78.6	125.1	188.8	190.0	239.2
	39.5	71.5	120.8	162.9	197.2	270.8
**F_a(i)_ ± κ_5%,3_ [N]**	38.3 ± 3.0	74.8 ± 9.6	129.4 ± 32.7	174.4 ± 36.6	196.9 ± 17.5	247.2 ± 59.8
**μ_(i)_** **[-]**	0.75	0.75	0.87	0.88	0.80	0.83

*^2a^ see Figure 6a, *^2b^ see Figure 6b, *^2c^ see Figure 6c, *^2d^ see Figure 6d, *^2e^ see Figure 6e, *^2f^ see Figure 6f.

**Table 4 sensors-23-03137-t004:** Measured force F_M1(i)_ [N] and F_M2(i)_ [N] for the weight of the load G_b(i)_ [N] at angle β = 20 deg.

G_b(i)_ [N]	51	100.1	149.1	198.1	247.2	296.2
**F_M1(i)_ [N]**	24.9	28.6	44.6	62.3	105.5	142.6
	25.8	29.1	42.7	61.1	106.4	137.3
	25.2 *^3a^	27.8 *^3b^	39.6 *^3c^	60.6 ^3d^	111.3 *^3e^	138.2 ^+3f^
**F_M2(i)_ [N]**	64.7	103.8	166.3	226.7	308.7	396.2
	65.6	103.5	162.6	217	301.2	390.3
	65.1 *^3a^	104.2 *^3b^	171.2 *^3c^	218.8 *^3d^	314.3 *^3e^	399.9 ^3f^
**F_M2(i)_ − F_M1(i)_ [N]**	39.8	75.2	121.7	164.4	203.2	253.6
	39.8	74.4	119.9	155.9	194.8	253.0
	39.9	76.4	131.6	158.2	203.0	261.7
**F_a(i)_ ± κ_5%,3_ [N]**	39.8 ± 0.2	75.3 ± 2.7	124.4 ± 18.2	159.5 ± 12.4	200.3 ± 14.0	256.1 ± 14.2
**μ_(i)_** **[-]**	0.78	0.75	0.83	0.81	0.81	0.86

*^3a^ see Figure 7a, *^3b^ see Figure 7b, *^3c^ see Figure 7c, *^3d^ see Figure 7d, *^3e^ see Figure 7e, *^3f^ see Figure 7f.

**Table 5 sensors-23-03137-t005:** Measured force F_M1(i)_ [N] and F_M2(i)_ [N] for the weight of the load G_b(i)_ [N] at angle β = 30 deg.

**G_b(i)_ [N]**	51	100.1	149.1	198.1	247.2	296.2
**F_M1(i)_ [N]**	25.9 *^4a^	41.1 *^4b^	63.8 *^4c^	98.7 *^4d^	132.3 *^4e^	167.9 *^4f^
	24.6	39.2	67.0	97.4	141.2	176.9
	24.2	39.8	69.7	89.9	132.4	163.6
**F_M2(i)_ [N]**	66.6 *^4a^	127.6 *^4b^	201.8 *^4c^	258.5 *^4d^	344.4 *^4e^	415.5 *^4f^
	65.3	122.6	207.3	269.1	352.8	398.8
	69.8	129.9	193.4	273.9	347.5	418.4
**F_M2(i)_ − F_M1(i)_ [N]**	40.7	86.5	138.0	159.8	212.1	247.6
	45.2	90.7	126.4	176.5	206.3	241.5
	45.6	90.1	123.7	184.0	215.1	254.8
**F_a(i)_ ± κ_5%,3_ [N]**	43.8 ± 7.9	89.1 ± 6.6	129.4 ± 21.9	173.4 ± 34.5	211.2 ± 12.3	248.0 ± 17.3
**μ_(i)_ [-]**	0.86	0.89	0.87	0.88	0.85	0.84

*^4a^ see Figure 8a, *^4b^ see Figure 8b, *^4c^ see Figure 8c, *^4d^ see Figure 8d, *^4e^ see Figure 8e, *^4f^ see Figure 8f.

**Table 6 sensors-23-03137-t006:** Measured force F_M1(i)_ [N] and F_M2(i)_ [N] for the weight of the load G_b(i)_ [N] at angle β = 40 deg.

**G_b(i)_ [N]**	51	100.1	149.1	198.1	247.2	296.2
**F_M1(i)_ [N]**	25.1	32.3	68.4	93.5	122.3	152.7
	24.9 *^5a^	29.3 *^5b^	71.4 *^5c^	95.4 *^5d^	127.7 *^5e^	146.3 *^5f^
	23.4	28.7	69.7	99.9	124.6	154.6
**F_M2(i)_ [N]**	65.4	105.3	195.7	262.7	324.1	395.9
	62.5 *^5a^	108.7 *^5b^	201.2 *^5c^	258.9 *^5d^	334.2 *^5e^	386.8 *^5f^
	63.9	107.4	204.7	264.2	325.1	392.5
**F_M2(i)_ − F_M1(i)_ [N]**	40.3	73	127.3	169.2	201.8	243.2
	37.6	79.4	129.8	163.5	206.5	240.5
	40.5	78.7	135	164.3	200.5	237.9
**F_a(i)_ ± κ_5%,3_ [N]**	39.5 ± 7.9	77.0 ± 6.6	130.7 ± 21.9	165.7 ± 34.5	202.9 ± 12.3	240.5 ± 17.3
**μ_(i)_ [-]**	0.77	0.77	0.88	0.84	0.82	0.81

*^5a^ see Figure 9a, *^5b^ see Figure 9b, *^5c^ see Figure 9c, *^5d^ see Figure 9d, *^5e^ see Figure 9e, *^5f^ see Figure 9f.

**Table 7 sensors-23-03137-t007:** Measured force F_M1(i)_ [N] and F_M2(i)_ [N] for the weight of the load G_b(i)_ [N] at angle β = 50 deg.

**G_b(i)_ [N]**	51	100.1	149.1	198.1	247.2	296.2
**F_M1(i)_ [N]**	23.9	32.1	62.7	97.2	131.3	154.1
	24.6	33.2	67.4	94.4	129.8	149.4
	25.2 *^6a^	35.8 *^6b^	66.1 *^6c^	90.8 *^6d^	134.1 *^6e^	152.6 *^6f^
**F_M2(i)_ [N]**	67.3	110.6	193.4	251.9	324.4	393.2
	62.2	109.9	183.8	250.8	323.3	392.3
	63.9 *^6a^	117.6 *^6b^	190.3 *^6c^	263.0 *^6d^	334.2 *^6e^	398.2 *^6f^
**F_M2(i)_ − F_M1(i)_ [N]**	43.4	78.5	130.7	154.7	193.1	239.1
	37.6	76.7	116.4	156.4	193.5	242.9
	38.7	81.8	124.2	172.2	200.1	245.6
**F_a(i)_ ± κ_5%,3_ [N]**	39.9 ± 8.9	79.0 ± 7.1	123.8 ± 18.7	161.1 ± 28.1	195.6 ± 11.5	242.5 ± 8.7
**μ_(i)_ [-]**	0.78	0.79	0.83	0.81	0.79	0.82

*^6a^ see Figure 10a, *^6b^ see Figure 10b, *^6c^ see Figure 10c, *^6d^ see Figure 10d, *^6e^ see Figure 10e, *^6f^ see Figure 10f.

## Data Availability

Not applicable.

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
