# Peer review of "A Laboratory Device Designed to Detect and Measure the Resistance Force of a Diagonal Conveyor Belt Plough"

_sensors, 2023, doi:10.3390/s23063137_

Round 1

Reviewer 1 Report

In this manuscript the amount of resistance generated by the conveyor belt diagonal plough is determined at various inclination angle. The experimental set up used in the study is well illustrated in the manuscript and the conclusions are derived from the results of the measurements.

The work presented in the paper is appropriate for publication.

Author Response

Dear reviewer of the article entitled "Laboratory equipment designed to detect and measure the resistance force of a diagonal plow of a conveyor belt".

Thank you for your review. We appreciate your willingness to review the article. We are very grateful for your review.

Sincerely, Hrabovský Leopold.

Reviewer 2 Report

The paper titled "A laboratory device designed to detect and measure the resistance force of a conveyor belt diagonal plough" presents a laboratory device designed to simulate a section of a conveyor belt with a diagonal plough installed. The aim of the study was to determine the amount of resistance generated by the conveyor belt diagonal plough placed at different angles of inclination β [deg] in relation to the longitudinal axis. The authors conducted experimental measurements in a laboratory and obtained time records of the tensile forces, from which they determined the magnitude of the force and the resistance during the ploughing operation provided by the diagonal plough. They also calculated the friction coefficient obtained during the movement of the diagonal plough when moving a piece of load with the defined weight from the working surface of the relevant conveyor belt. The findings suggest that the specific movement resistance of the conveyor belt diagonal plough is 0.33 [N·N-1], and the maximum value of the arithmetic mean for the friction coefficient in motion µ = 0.86 was measured at the inclination angle β = 30 deg of the diagonal plough. Overall, this paper provides valuable insights into the resistance generated by the conveyor belt diagonal plough and its impact on the conveyor belt movement, which can be beneficial for researchers and engineers working in this field.

However, the following are my comments, which can help authors to further improve the manuscript, if properly incorporated.

1.     During them, a plastic crate, representing a piece load, transported on the surface of a conveyor belt at a  constant speed came into contact with the front surface of a conveyor belt diagonal plough’…. Sentence is incorrect and should be corrected.

2.     The figure 1, is taken form ref[17]. I think, there seems no significant need to add this figure. In fact, discussion about in the introduction may work well.

3.     Figure 4 is unnecessary, and very much generic. Simple discussion with reference will be enough.

4.      I will highly recommend to add a comparison table which can provide a comprehensive comparison between the presented work with that of previous studies.

5.      There are many grammatical mistakes, which should be corrected in the revised version.

6.      A video demonstration should also be added, this will help readers to get clear understanding of the work presented.

7.      Please add more reference in the introduction part with their appropriate discussion.

Author Response

Dear reviewer of the manuscript,

Thank you for your review. We are very grateful for your review.

Answer to point 1:

We have replaced the original text in the manuscript (see p. 1) with this: "IDuring them, a plastic storage box, representing a piece load, transported on the surface of a conveyor belt at a constant speed came into contact with the front surface of a conveyor belt diagonal plough."

Answer to point 2:

In the description of Fig. 1, which was taken from reference [17], we removed the required data.

Answer to point 3:

I removed the reference [] in the description of Fig. 4.,

TI inserted this comment into the text of the post "During laboratory measurements, a load sensor cable 26 equipped with a D-Sub plug was plugged into the socket of the BR5D measuring module [4] of the DS NET strain gauge. The notebook was connected to the DS NET strain gauge [26] using a network cable with connectors RJ-45 on both ends.

Answer to point 4:

Dear, from the analysis of the articles related to the compaction plow, we found only those articles that are listed in the list of references, on the recommendation of the opponent #2.

- Hrabovský, L.; Kulka, J.; Mantic, M. Experimental Expression of Belt Conveyor Plow Resistance. MATEC Web of Conferences 2019, 263, p. 01006.

- Shah K. P. Construction and Maintenance of Belt Conveyors for Coal and Bulk Material Handling Equipment.

- Nata, Y.; Mulyana, Y.; Haris, O.; Hidayat, M. S. Belt Conveyor V-Flow Model Optimization in Obtaining Optimum Coal Flow Through CFD. In 2018 International Conference on Computing, Engineering and Design (ICCED), Bangkok, Thailand, 6-8 September 2018.

- Efremenkov, V.V. Details on the use of diagonal plows in batch and chip conveyor lines. Glass Ceram 2020, 76, pp. 391–395.

- Moor, B. S. Belted Vs. chain lamellar bagasse conveyors for feeding boilers. Proc South African Sugar Technol Assoc 2020, 1, pp. 285-289.

- Makutu, M. C., & von Kallon, D. V. Method of operating a coal mining plow and tiler. Proceedings of the 3rd Asia Pacific International Conference on Industrial Engineering and Operations Management 2022, pp. 13-15.

Answer to point 5:

The text of the contribution was submitted for assessment by a university English language teacher (see https://www.vsb.cz/personCards/personCard.jsp?lang=en&person=chu01), who corrected the text.

Answer to point 6:

I am attaching a set of photos from the actual measurement in a file named "*.pdf".

Answer to point 7:

Dear reviewers,
analysis of articles related to the diagonal plow, we found only the following articles from the article search for your recommendation:
I would like to ask for your indulgence in extending further links in the introductory section to the relevant discussion. Linking would not apply to articles dealing with diagonal plows.

Thank you very much, Hrabovský Leopold.

Reviewer 3 Report

Dear Authors,

the Article is relevant and scientific-sounding. However, I would like to point out two notes:

1. The Discussion section should not contain figures, tables, they should enforce the Result section. So I suggest discussing the future development of research, its constrains and main advantages. 

2. The References list should be updated with new and international literature.

Good luck!

Author Response

Hello, I would like to thank you for your review of our manuscript.

1) Dear reviewer,
I corrected the "Discussion" chapter according to the recommendation of opponent No. 2. Specifically - I moved Fig. 11 to Fig. 13 to chapter 3. Results.

2) Dear reviewer,
from the performed analysis of articles related to the compaction plow, we found only those articles that are listed in the list of used literature, to the recommendation of opponent No. 2.
- Hrabovsky, L.; Kulka, J.; Mantic, M. Experimental expression of the resistance of belt conveyor’s plough. MATEC Web of Conferences 2019, 263, p. 01006.
- Shah K. P. Construction and Maintenance of Belt Conveyors for Coal and Bulk Material Handling Plants. 
- Nata, Y.; Mulyana, Y.; Haris, O.; Hidayat, M. S. Optimizing the V Flow Model on the Belt Conveyor in Getting the Optimal Coal Flow Rate with CFD Method. In 2018 International Conference on Computing, Engineering, and Design (ICCED), Bangkok, Thailand, 6-8 September 2018.
- Efremenkov, V. V. Particulars of Using Diagonal Plows in Batch and Cullet Conveyor Lines. Glass Ceram 2020, 76, pp. 391–395. 
- Moor, B. S. Belt vs. chain-slat bagasse conveyors for boiler feeding. Proc South African Sugar Technol Assoc 2020, 1, pp. 285-289.
- Makutu, M. C., & Von Kallon, D. V. Method of Operation of a Plough and Tiling Device Designed for Coal Mining. Proceedings of the 3rd Asia Pacific International Conference on Industrial Engineering and Operations Management 2022, pp. 13-15.
3nd reviewer

Thank you, with respectful regards, Hrabovský Leopold

Reviewer 4 Report

1. This article is more like an experiment record, hard to find any scientific sound.

2. Many details havent been mentioned. What is the target of the research, which kind problems wanted to solve, how accurate the result required? Without these basic information make the reading become vague. 

Author Response

Hello, I would like to thank you for your review of our manuscript.

The aim of the article is to obtain time records obtained by measuring tensile forces, from which it is possible to determine the magnitude of the force. It provides the plowing resistance that a diagonal plow provides when acting on a piece load that is placed on the working surface of the conveyor belt. From the measured values of the tensile forces entered in the tables, this contribution presents the calculated values of the coefficient of friction obtained during the movement of the diagonal plow when moving a piece of load of a defined weight from the working surface of the respective conveyor belt.

The laboratory device described in the article enables us to measure the magnitude of tensile force, which is generated as a resistive force when trying to set the non-moving conveyor belt in motion and also when it moves at a constant speed on the conveyor idlers, supporting this belt.

One end of a steel rope is wound onto the drive drum and the other end is mechanically attached to the load cell and the front area of the conveyor belt produces a tensile force. When the tensile force reaches the magnitude of the force needed to spin the idlers supporting the conveyor belt, the moment comes when the conveyor belt is set in motion from the standstill position. The time recording of the measured tensile force obtained using DEWESoft X2 SP5 software allows you to determine the instantaneous value of the tensile force, at any time of the recorded measurement, during the start-up and movement of the conveyor belt.

Thank you, with respectful regards, Hrabovský Leopold

Round 2

Reviewer 3 Report

Good luck!

Reviewer 4 Report

No more